# Tails of Instability and Decay: a Hydrodynamic Perspective

Olalla A. Castro-Alvaredo$^\heartsuit$, Cecilia De Fazio$^\diamondsuit$, Benjamin Doyon$^\spadesuit$, and Aleksandra Ziolkowska$^\clubsuit$

$^{\heartsuit\diamondsuit}$ Department of Mathematics, City, University of London, 10 Northampton Square EC1V 0HB, UK

$^\spadesuit$ Department of Mathematics, King's College London, Strand WC2R 2LS, UK

$^\clubsuit$ Rudolf Peierls Centre for Theoretical Physics, Parks Road, Oxford OX1 3PU, UK

In the context of quantum field theory (QFT), unstable particles are associated with complex-valued poles of two-body scattering matrices in the unphysical sheet of rapidity space. The Breit-Wigner formula relates this pole to the mass and life-time of the particle, observed in scattering events. In this paper, we uncover new, dynamical signatures of unstable excitations and show that they have a strong effect on the non-equilibrium properties of QFT. Focusing on a 1+1D integrable model, and using the theory of Generalized Hydrodynamics, we study the formation and decay of unstable particles by analysing the release of hot matter into a low-temperature environment. We observe the formation of tails and the decay of the emitted nonlinear waves, in sharp contrast to the situation without unstable excitations. We also uncover a new phenomenon by which a wave of a stable population of unstable particles may persist without decay for long times. We expect these signatures of the presence of unstable particles to have a large degree of universality. Our study shows that the out-of-equilibrium dynamics of many-body systems can be strongly affected not only by the spectrum, but also by excitations with finite life-times.

**Keywords:** Out-of-Equilibrium Dynamics, Integrability, Generalized Hydrodynamics, Unstable Particles

$^\heartsuit$ o.castro-alvaredo@city.ac.uk
$^\diamondsuit$ cecilia.de-fazio.2@city.ac.uk
$^\spadesuit$ benjamin.doyon@kcl.ac.uk
$^\clubsuit$ aleksandra.ziolkowska@physics.ox.ac.uk

September 1, 2021

# 1 Introduction

Physics far from equilibrium has received a large amount of interest recently. It is now understood that non-equilibrium dynamics offers a powerful new way of studying strongly correlated many-body systems, as it brings out properties that can be hidden in equilibrium situations. This viewpoint has gained strong traction, particularly in one dimension [1, 2, 3, 4, 5, 6, 7, 8]. It is interesting to apply this principle to quantum field theory (QFT) far from equilibrium. A paradigm is that the essence of quantum dynamics in QFT is accessed by scattering theory, describing how few particles interact [9, 10, 11]. But scattering events – perhaps the simplest example of non-equilibrium dynamics – mainly teach us about the vacuum state, and what happens at low densities. Further, in the context of applications to many-body systems, scattering experiments are difficult to implement. Dynamics with finite energy and spatial particle densities can be argued to be more common and accessible, and to have a richer physics.

A case in point is the spectrum of asymptotic particles. This is a basic ingredient of scattering theory, but it is not sufficient in order to fully understand finite-density physics. Other emergent structures of QFT may play an important role. One example is the phenomenon of confinement, which taps into the internal structure of asymptotic particles. As found in [12, 13, 14] this and other qualitatively similar [15] phenomena can be linked quite explicitly to the non-equilibrium dynamics that follows so-called quantum quenches [16, 17]. Another phenomenon, only indirectly observed in scattering events, is the existence of particles with finite life-times, which are absent from the asymptotic spectrum. Technically, they appear as poles in the unphysical sheet of the analytically continued scattering phase function [11]. The Breit-Wigner formula allows one to evaluate their mass and life-time from the position of this pole [18]. At nonzero temperatures, physical intuition suggests that unstable particles are constantly formed and destroyed, and thus truly exist in finite proportions. As such, they may be expected to have a strong effect on the dynamics far from equilibrium at finite densities.

In this paper, we argue that this is the case, by studying the propagation of nonlinear waves at finite energy densities. We focus on a 1+1D integrable QFT, which admits one unstable particle interpreted as a loosely bound state of two different asymptotic particle types. We analyse the setup where a high-temperature finite region, with enough energy to form a large number of unstable particles, is released into a colder environment, with few unstable particles. We study the emitted waves and their large-time behaviour by numerically solving the associated hydrodynamic equations. At the energies considered, asymptotic particles have velocities extremely near to the speed of light (set to $c = 1$). Then, the standard picture, without unstable particles, is that after the splitting of the initial high-temperature region, long-lived, well-localised regions of higher energy density propagate at unit velocity in both directions (if the colder environment is the vacuum, over extremely long times, the waves would slowly decompose into their asymptotic particle content). We argue that the formation, propagation and decay of unstable particles drastically modifies this dynamics. Three main observations are made: (a) the emitted waves *partially decay* after the splitting of the initial high-temperature region; (b) long tails trail the well localised high-density waves; and (c) in an environment with nonzero but low temperature, higher waves persist, seemingly indefinitely.

We interpret all observations using the physics of unstable particles. We show that observations (a) and (b) are due to the decay of unstable particles as the waves propagate. This

is the expected physics, of which our non-equilibrium finite-density dynamics offer a novel and perhaps more direct observation. Observation (c) is associated to a new phenomenon hitherto not observed: although the low-temperature bath does not support unstable particles, due to subtle interaction effects, residual *stable* waves of unstable particles emerge. These may play the role of stable excitations in an effective scattering theory at finite densities.

This paper is organized as follows: in Section 2 we briefly introduce the model and the generalized hydrodynamic description of the problem. In Section 3 we present and interpret the main numerical results of the paper. In Section 4 we provide further details on the structure of the subsidiary peak, focussing on its distribution of effective and propagation velocities. In Section 5 we discuss in further detail the stark differences between our results and those for a free theory, particularly when analyzing the particle densities. We conclude in Section 6. In Appendix A we review some features of the numerical algorithm.

## 2   Model and Hydrodynamic Equations

1+1D quantum integrable models have found many applications over the past two decades. Besides having a rich underlying mathematical structures, they play a role in various areas of modern theoretical physics: from the emergence of integrability in string/gauge theories [19], to the experimental realisation of quantum integrability [20, 21] and the measurement of quantities that directly compared to analytic formulae [22, 23]. The investigation of the out-of-equilibrium dynamics of integrable models, largely spurred by the Quantum Newton's Cradle experiment [24], has become an active area of research [25]. In particular, generalised hydrodynamics (GHD) [26, 27], the hydrodynamic theory which accounts for generalized thermalisation (GGE) [28] in quantum integrable models, has been very successful (see e.g. [8] for a recent review).

Integrability implies that multi-particle scattering is elastic and factorises into two-body events [29, 30, 31]. Interestingly, these constraints do not forbid the existence of unstable bound states even if few theories with this feature are known. In this paper, we use the GHD approach to study a model whose two-body $S$-matrix has a pole in the rapidity's unphysical sheet, which, as mentioned, implies the presence of an unstable particle. In a previous work [32] it was found that the presence of this unstable excitation gives distinctive features of various hydrodynamic quantities in stationary states. Here, using GHD and these results, we obtain a direct observation of the physics of unstable particles in real space, including their decay and long-time persistence.

We study the $SU(3)_2$-homogenous sine-Gordon (HSG) model. It has a spectrum of two self-conjugate particles, and a $S$-matrix that breaks parity invariance. It is the simplest of the large family of $G_k$-HSG models labelled by a simply-laced algebra $G$ and an integer index $k$ (the level), whose integrability and $S$-matrices were studied in [33, 34, 35, 36]. Much work was devoted to the computation of form factors [37, 38, 39, 40], the application of the thermodynamic Bethe ansatz [41, 42, 43], and the study of mass-coupling relations [44, 45]. The GHD necessitates the two-body scattering phases of the model:

$$\varphi_{\pm\pm}(\theta) = 0, \qquad \varphi_{\pm\mp}(\theta) = \mathrm{sech}(\theta \pm \sigma), \quad \sigma \in \mathbb{R} \tag{1}$$

where $\theta$ is the rapidity difference and $\pm$ label the particle species. Particles $\pm$ have equal mass

$m$. The Breit-Wigner formula predicts the formation of an unstable particle with mass and decay width both proportional to $e^{|\sigma|/2}$ for $|\sigma| \gg 0$. For $\sigma$ positive the scattering phases dictate that a particle $\pm$ has maximum interaction with a particle $\mp$ if it hits it from the right/left and their rapidity difference is of order $\theta = \mp\sigma$. This feature means that interaction for each particle is maximised for either positive or negative rapidities, whereas particles behave as free fermions otherwise. Particles of the same kind interact as free fermions. Thus, in a Gibbs ensemble at temperature $1 \ll T \ll e^{|\sigma|/2}$ the theory flows to the conformal field theory of central charge $c = 1$ for two free fermions, whereas for $T \gg e^{|\sigma|/2}$ it flows to an interacting theory with central charge $c = 6/5$ [33]. The larger number of degrees of freedom at high temperatures is linked to the presence of a finite proportion of unstable particles, allowed by the large energy.

We choose an initial Gaussian temperature profile

$$T(x) = T_a + (T_m - T_a)e^{-x^2} \quad \text{with} \quad T_a, T_m \in \mathbb{R}_{\geqslant 0}, \tag{2}$$

where $x$ is the space coordinate. We will refer to $T_a$ as the bath temperature. Given such an initial configuration, the dynamics can be obtained by GHD as described for instance in [46, 47, 48, 49]. In GHD, asymptotic particles, characterised by their rapidity $\theta$ and quantum number $b$, are "dressed", within finite density states, into fluid modes. The state at any space-time point $x, t$ is then described by an occupation function $n^b(x, t, \theta)$. The thermodynamic Bethe ansatz [52] allows us to evaluate the initial condition $n^b(x, 0, \theta)$ from (2) using the local density approximation. For our analysis, the three quantities of interest are the spectral density of particles $\rho_p^\pm(x, t, \theta)$, the spatial particle density $\mathsf{q}_0^\pm(x, t)$, and the effective velocity $v^{\text{eff},\pm}(x, t, \theta)$, which is the fluid velocity of mode $\theta, \pm$. They are defined as

$$\rho_p^\pm(x, t, \theta) = e^{\text{dr},\pm}(x, t, \theta)n^\pm(x, t, \theta), \tag{3}$$

$$\mathsf{q}_0^\pm(x, t) = \int_{-\infty}^\infty \frac{d\theta}{2\pi}\rho_p^\pm(x, t, \theta), \tag{4}$$

$$v^{\text{eff},\pm}(x, t, \theta) = p^{\text{dr},\pm}(x, t, \theta)/e^{\text{dr},\pm}(x, t, \theta). \tag{5}$$

Here $e^\pm(\theta) = \cosh\theta$ is the energy, and $p^\pm(\theta) = \sinh\theta$ the momentum (the mass is set to unity). The dressing operation "dr" is defined by the equations

$$\sum_{b=\pm} \int_{-\infty}^\infty \frac{d\alpha}{2\pi} G_b^\pm(\theta, \alpha)h^{\text{dr},b}(x, t, \alpha) = h^\pm(\theta), \tag{6}$$

with

$$G_b^\pm(\theta_2, \theta_1) = 2\pi\delta(\theta_{12})\delta_{b\pm} - \varphi_{\pm\mp}(\theta_{12})n^b(x, t, \theta_1)\delta_{b\mp}, \tag{7}$$

and $\theta_{12} := \theta_1 - \theta_2$. The occupation functions evolve according to the GHD equation [26, 27]

$$\partial_t n^\pm(x, t, \theta) + v^{\text{eff},\pm}(x, t, \theta)\partial_x n^\pm(x, t, \theta) = 0. \tag{8}$$

In free theories $v^{\text{eff}}(x, t, \theta) = \tanh\theta$, and therefore, at large temperatures, where large rapidities are involved, excitations mostly move at velocities $\pm 1$. Typically, interaction in most integrable QFTs does not qualitatively change the effective velocity, and the same holds near interacting conformal points. This leads to the splitting of the original density maximum at $x = 0$ into two identical maxima propagating in opposite directions [46, 47, 48, 49]. This is in fact also the phenomenology of non-integrable models, where the dynamics is dominated by two "sound modes". We will see that the presence of unstable particles modifies this picture substantially.

# 3  Numerical Results

In order to obtain numerical results, we adapted the iFluid package [53], a MATLAB code that solves GHD for a wide range of conditions and models. More details are presented in Appendix A. We choose $\sigma = 10$: for temperatures $2T \lesssim e^5$ the model describes two free Majorana fermions, at higher temperatures unstable particles are formed. In order to best observe this threshold, the bath temperature $T_a$ is either 0 or within the free fermion regime $2T_a = e^3$, and the maximum temperature $2T_m = e^7$ is within the interacting region. As emphasised in [32], the analysis of spectral densities is helpful in determining the composition of finite-density states. We note that the hydrodynamic approach is valid for typical inter-particle distances ($d$) and interacting lengths ($d'$) much smaller than variation lengths ($d''$). With the chosen parameters, $d = O(10^{-2})$, $d' = \max(\varphi(\cdot))/p'(\theta) = O(10^{-1}) - O(10^{-3})$ for relevant values of rapidities, and $d'' = O(1)$ (see FIG. 1), we indeed have $d, d' \ll d''$.

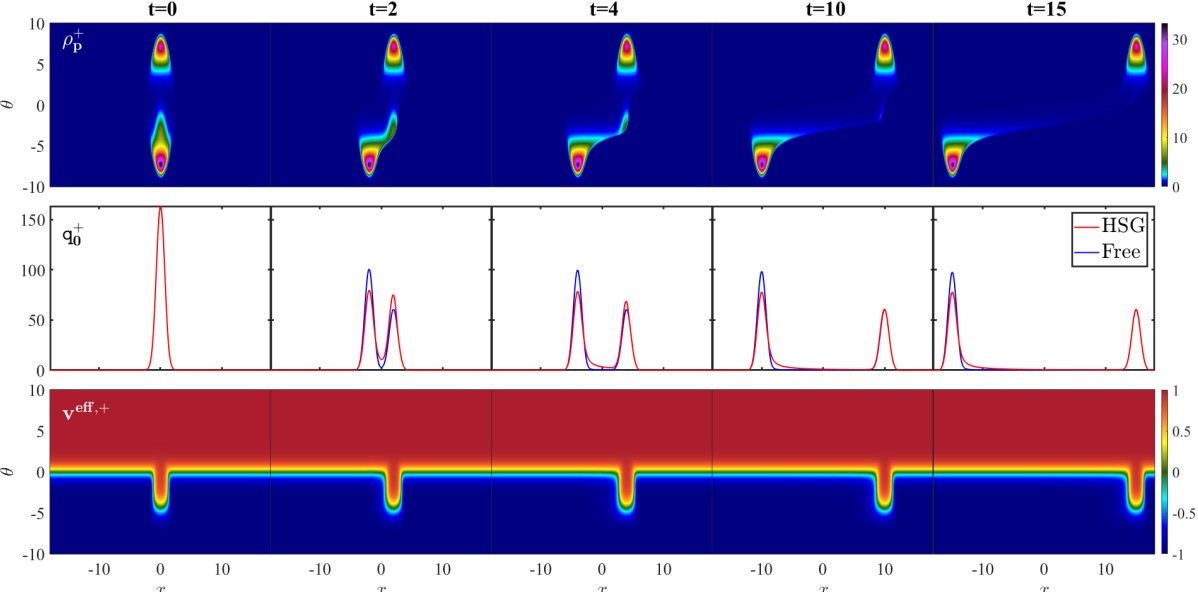

Figure 1: Particle + dynamics for various time snapshots. The parameters are $\sigma = 10$, $T_a = 0$ (no bath) and $2T_m = e^7$. Row 1: Spectral density exhibiting three characteristic local maxima, the least of which decays in time leading to the formation of a tail. Row 2: Spatial particle density exhibiting both a tail and decay (red) and its freely-evolved version (blue) exhibiting neither of the two. Row 3: Effective velocities dominated by the values $\pm 1$ but exhibiting an intermediate plateau in rapidity space which is correlated with the decaying peak of the spectral density. The dynamics of particle $-$ follows from the relations: $\rho_p^+(x, t, \theta) = \rho_p^-(-x, t, -\theta)$, $\mathsf{q}_0^+(x, t) = \mathsf{q}_0^-(-x, t)$ and $v^{\text{eff},+}(x, t, \theta) = -v^{\text{eff},-}(-x, t, -\theta)$.

Consider a zero-temperature environment, FIG. 1. The first row illustrates the behaviour of the spectral density of particle + as a function of time. At time $t = 0$ we observe the equilibrium features first discussed in [32]: three local maxima in rapidity space centred around $\theta \approx \pm \log(2T_m) = \pm 7$ and $\theta \approx \log(2T_m) - \sigma = -3$. The peak around $\theta \approx 7$ (free fermion peak)

in not interacting and has the free fermion shape for temperature $T_m$; the opposite maximum at $\theta \approx -7$ (interacting peak) is higher; the peak around $\theta \approx -3$ (subsidiary peak) is a consequence of interaction with the interacting peak of particle $-$ (it would not be present for free fermions). The subsidiary peak represents the proportion of particles $+$ loosely involved in a bind with particles $-$: these are the unstable particles. The effective velocity (the third row), for $t = 0$, shows that the interacting (free fermion) peak moves at speed $-1$ ($+1$), and the subsidiary peak contains a spectrum effective velocities captured by the "rainbow" colouring and the presence of an intermediate plateau, see also FIG. 4 in Section 4. The particle $+$'s subsidiary peak's velocities are pushed upwards as they form unstable but coherent bounds with particles $-$.

Turning on time, new features emerge not seen previously. The most salient feature is that the subsidiary peak moves as per the initial splitting, but then falls apart, leaving behind a tail of particles with velocities tending to $-1$. Unstable particles start moving at speed near $+1$, but, as they enter the zero-temperature environment, decay faster than they can form, and we see the particle $+$ components un-binding and recovering their non-interacting speeds near to $-1$. The effect can be seen directly in the spatial density, as shown in the second row of FIG. 1. After the initial splitting, the right-moving wave reduces in time, leaving just the free fermion contribution (which remains unchanged for larger times) and a tail attached to the left-moving wave at $x < 0$. Comparison with what happens for free-particle evolution from the same initial distribution reveals starkly that this is the physics of unstable particles, see also Section 5. With free evolution two stable waves are emitted without decay as expected; the right-moving wave agrees with the final stage of the decaying wave of the interacting evolution, while the left-moving one is higher, as it contains particles that would otherwise lie in the tail. Clearly, there is not enough dispersion in the original distribution of rapidities to explain the decay and formation of tails: the interacting non-equilibrium dynamics, whereby unstable particles decompose as they enter the low-temperature bath, is by far the dominant effect.

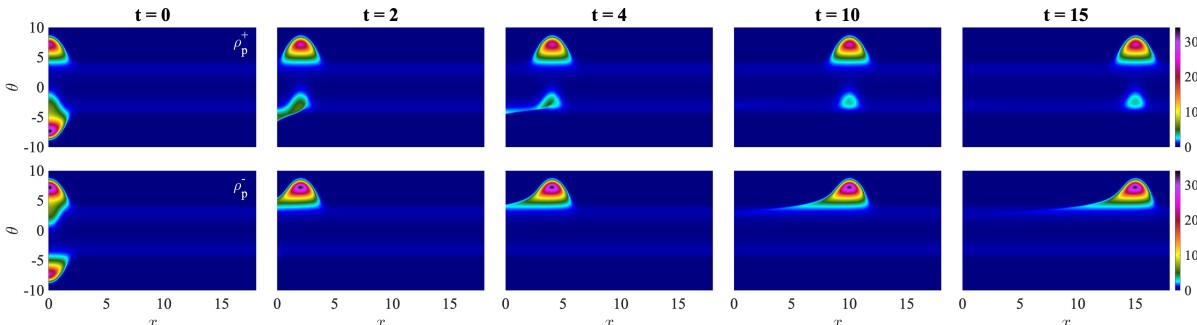

Figure 2: Snapshots of the spectral densities $\rho_p^{\pm}(x, t, \theta)$ for $x \geqslant 0$ in the presence of a bath at temperature $2T_a = e^3$, $\sigma = 10$ and $2T_m = e^7$. Recall that $\rho_p^+(x, t, \theta) = \rho_p^-(-x, t, -\theta)$. Whilst the interacting and free fermion peaks remain largely unchanged, the bath facilitates the formation of a persistent peak that travels at speed $+1$ "riding" on the bath. The two light-blue ridges are also due to the bath whose temperature is in the free fermion region of the theory. The ridges look "static" because they are uniformly distributed in space but represent particles propagating with opposite effective velocities $\pm 1$. See also the videos [51, 50].

Do these behaviours change in the presence of a bath at nonzero temperature? FIG. 2 shows the $x \geqslant 0$ part of the functions $\rho_p^{\pm}(x, t, \theta)$ (see also the video [50]). Two main changes occur. First, two continuous, static ridges are formed centred around $\theta \approx \pm \log(2T_a) = \pm 3$: the free fermion equilibrium distribution for temperature $T_a$. Second, most importantly, the subsidiary peak observed at time $t = 0$ no longer fully disintegrates under time evolution. Instead, it largely persists, propagating on top of the $\theta = -3$ ridge. Because of the spread of effective velocities in the initial subsidiary peak, this cannot be explained by a large population of particles at constant velocity $+1$. Indeed, there is a large difference between the effective velocity and the propagation velocity of these particles, as discussed in Section 4. Why is this wave travelling at speed $+1$, riding on the bath? The answer is that the large wave of particles $-$ going at velocity $+1$ interacts with particles $+$ in the bath, because they are present around the rapidities of the subsidiary peak. They form unstable particles as they pass by, thus changing the bath density. This is akin to having a fluid that is magnetic, and running a magnet pass it. We see a wave that follows the magnet but the fluid itself does not need to move. This is a hydrodynamic effect never previously seen in the GHD context. We point out that the life-time of the unstable particles is of order $e^{-\frac{\sigma}{2}}$: unstable particles rapidly decay and reform within the persistent peak, with the end result of preserving it for large times. Comparing with free-particle evolution, where no persistent wave occurs, gives further insight, as discussed in Section 5.

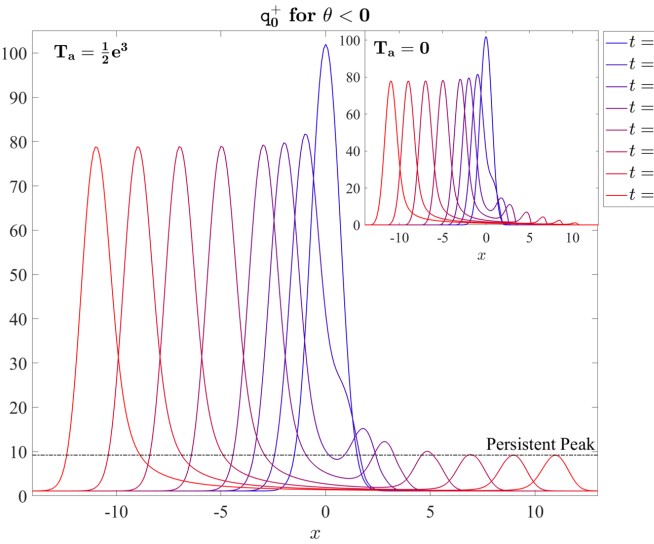

Figure 3: The "partial" spatial particle density $\mathsf{q}_0^+(x, t)$ obtained from integrating $\rho_p^+(x, t, \theta)$ for $\theta < 0$ showing the decay (no bath, inset) and persistence (bath, main figure) of the subsidiary peak. See also the video [51].

The effect of the bath is seen starkly in FIG. 3 (see also the video [51]): we plot the "partial" spatial particle density for particle $+$, integrating the spectral density for $\theta < 0$. This is equivalent to subtracting the free-fermion contribution, which remains unchanged over time. Both the decay of the subsidiary peak in the absence of a bath (inset) and its persistence in the presence of a bath (main figure) are made evident. It is worth mentioning that the presence of a persisting peak is not a consequence of fine tuning of parameters but a robust effect, present for $\sigma > T_m > \frac{\sigma}{2} > T_a > 0$, as peaks have finite extension and will overlap for a wide range of temperatures, see [54].

Further evidence that the signatures of decay found here provide a non-trivial characterisa-

tion of unstable particles is provided in Sections 4 and 5.

# 4    A Closer Look at the Subsidiary Peak

We have seen that both the partial spatial particle density (see FIG. 3) and the spectral density (see FIGs. 1 and 2) exhibit a peak which either decays or persists for large times, depending on whether or not there is a bath. In this section, we examine the features of this peak in a bit more detail. A more complete analysis will be presented elsewhere [54]. For now, we will look at cross-sections of the spectral density's subsidiary peak and of the velocity distribution for particles in this peak, for fixed values of $x$ and $t$. We will make an important distinction between effective and propagation velocities and show that these are in general different, a feature that is dictated by interaction with the interacting peak of the opposite particle type and would therefore be absent if evolution was free, as explored in Section 5.

## 4.1    Effective vs. Propagation Velocities

Let us consider the effective velocity of particles both in the presence and in the absence of a bath. We will report on particle $+$ with particle $-$ admitting an analogous analysis. Data facilitating this discussion is presented in Table 4.

| | No bath | | | | | Bath | | | | |
|---|---|---|---|---|---|---|---|---|---|---|
| $t$ | $x^*$ | $\theta^*$ | $h^*$ | $x^*/t$ | $v^*$ | $x^*$ | $\theta^*$ | $h^*$ | $x^*/t$ | $v^*$ |
| 2.00 | 1.90 | -3.30 | 5.89 | 0.95 | 0.61 | 1.80 | -3.44 | 5.89 | 0.90 | 0.54 |
| 3.00 | 2.80 | -3.15 | 5.50 | 0.93 | 0.65 | 2.80 | -3.07 | 5.47 | 0.93 | 0.67 |
| 4.00 | 3.80 | -2.70 | 5.07 | 0.95 | 0.76 | 3.80 | -2.70 | 5.03 | 0.95 | 0.76 |
| 5.00 | 4.90 | -2.17 | 4.80 | 0.98 | 0.84 | 4.70 | -2.54 | 4.53 | 0.94 | 0.77 |
| 6.00 | 5.70 | -2.17 | 4.57 | 0.95 | 0.82 | 5.70 | -2.24 | 4.06 | 0.95 | 0.81 |
| 7.00 | 6.50 | -2.17 | 3.78 | 0.93 | 0.78 | 6.90 | -2.39 | 3.42 | 0.99 | 0.81 |
| 8.00 | 7.50 | -1.79 | 2.90 | 0.94 | 0.81 | 7.90 | -2.39 | 3.41 | 0.99 | 0.81 |
| 9.00 | 8.40 | -1.64 | 2.19 | 0.93 | 0.79 | 9.00 | -2.39 | 3.41 | 1.00 | 0.81 |
| 10.00 | 9.40 | -1.26 | 1.50 | 0.94 | 0.80 | 10.00 | -2.39 | 3.41 | 1.00 | 0.81 |
| 11.00 | 10.30 | -1.03 | 0.96 | 0.94 | 0.77 | 11.00 | -2.39 | 3.42 | 1.00 | 0.81 |
| 12.00 | 11.30 | -0.50 | 0.56 | 0.94 | 0.78 | 12.00 | -2.39 | 3.42 | 1.00 | 0.81 |

Table 1: Numerical results for the subsidiary peak of the spectral density of particle $+$ in the absence ($T_a = 0$) and presence ($2T_a = e^3$) of a bath. The other parameters are fixed to $\sigma = 10$ and $T_m = 2e^7$ as before. $x^*$, $\theta^*$ and $h^*$ are the phase-space coordinates and the height of the local maximum of the subsidiary peak, respectively and $v^* \equiv v^{\text{eff},+}(x^*, t, \theta^*)$. The quantity $x^*/t$ represents the observed propagation velocity of the subsidiary peak of particle $+$ resulting from its interaction with the interacting peak of particle $-$ which propagates at speed $+1$. The numerical uncertainty of phase-space coordinates is $\Delta x^* = 0.10$ and $\Delta \theta^* = 0.07$ to $0.08$, with lower rapidity resolution for less negative values of $\theta^*$ (as discussed in Appendix A).

We consider two different velocities: the standard effective velocity $v^* := v^{\text{eff}}(x^*, t, \theta^*)$ evaluated at the maximum of the subsidiary peak, and the *propagation* velocity, $x^*/t$ that is the actual speed at which the maximum of the peak is seen to propagate. The choices of times $t \in [2, 12]$

was dictated by the time of clear separation of the subsidiary peak from the interacting peak at $t = 2$ and the time at which almost complete decay of the subsidiary peak is observed for no bath at $t = 12$.

Looking at this table it is perhaps most striking that the effective and propagation velocities are markedly different both in the presence and in the absence of a bath. Thus, even if the presence of a bath gives rise to the magnetic fluid phenomenon described in the paper, that is, the emergence of a persisting peak that seems to ride on top of the bath, the fact is that, in the absence of a bath, the interaction with particles of type $-$ still has a huge influence on the propagation velocity of the subsidiary peak. Indeed, the peak propagates faster than would be expected from its effective velocity distribution and this can only be attributed to non-trivial interaction with particles of type $-$ which is inextricably linked to the presence of unstable particles. This increase in velocity is however larger in the presence of a bath and for large times it reaches the maximum value $+1$.

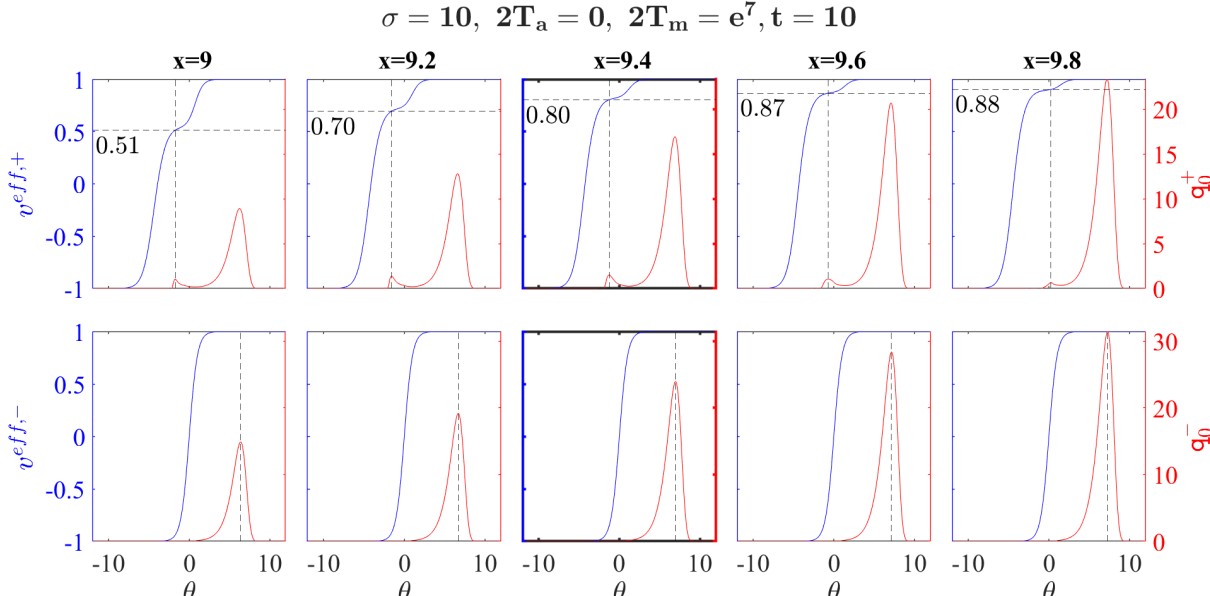

Figure 4: Effective velocities (left axis, blue) and spectral density cross-sections (right axis, red) of particle $+$ in row 1 and $-$ in row 2 at time $t = 10$ in the case of no bath. For particle $+$ we see the subsidiary (decaying) and free fermion peaks whereas for particle $-$ we see the interacting peak. Columns show these quantities sequentially for positions $x = 9, 9.2, 9.4, 9.6, 9.8$. The central emphasised column corresponds to $x = x^* = 9.4$, that is the position of the maximum of the subsidiary peak of particle $+$. The vertical dashed lines indicate the maximum of particle $+$ subsidiary peak ($\theta^* = -1.26$) in row 1 and the maximum of particle $-$ interacting peak in row 2, which is around the value $\theta^* + \sigma$ as dictated by the structure of the scattering phases. The horizontal dashed lines indicate the effective velocity of the maximum of particle $+$ subsidiary peak and are annotated with the corresponding numerical values.

In addition, we can make the following interesting observations:

- For early times data are very similar both in the presence and absence of a bath. Indeed, for times $t \in [2, 7]$ decay occurs whether or not there is a bath (a phenomenon we have not discussed in the paper). We can see this decay most clearly from the table and the height of maximum $h^*$ which reduces from 5.89 at $t = 2$ to 3.42 at $t = 7$. The way in which this decay or particle loss occurs is asymmetric in $\theta$ and results in an "under-cut" in the shape of the subsidiary peak, with the lost particles contributing to a growing tail linking the subsidiary peak to the interacting peak of the same particle species (this tail is seen most clearly in FIG. 1 of the paper). As a result, not only the height of the maximum is reduced but its position in phase-space shifts towards less negative values of $\theta$.

- In the absence of a bath, further asymmetric decay of the subsidiary peak continues until the peak disintegrates first into a front-like feature and ultimately into a section of the tail joining the free fermion and the interacting peak of the same species. It needs to be emphasised that the decay rate cannot be accounted for by the velocity profile alone. Since the propagation and the effective velocity are different, we know that interactions must play a fundamental role in this decay process.

- In the presence of a bath, the subsidiary peak becomes persistent from time $t = 7$, with the rapidity space position fixed at $\theta^* = -2.39$, height $h^* = 3.42$ and propagation velocity $x^*/t = 1$. The fact that the subsidiary peak moves along the interacting peak at velocity $+1$ rather than the much slower effective velocity of its constituents is the embodiment of the magnetic-fluid mechanism. The subsidiary peak does not consist of the same particles propagating but instead of newly formed bound-states between the particles of the interacting peak of particle $-$ and the particle reservoir of particle $+$ available from the bath ridge. Nevertheless, the asymmetric decay process, being independent from the persistent feature, is still present in the case with bath. It is only masked by the dominant mechanism of the magnetic-fluid for times larger than $t = 7$.

## 4.2 On the Shape of the Subsidiary Peak

For time $t = 10$, a deeper understanding of the data can be reached in conjunction with FIGs. 4 and 5 which show cross-sections of the effective velocity and spectral density of particles $\pm$ for the bath and no bath situations. In particular, we focus here on the shape of the subsidiary peak which can now be accessed more clearly through cross-sections at fixed space-time positions. Let us summarize our main observations:

- In FIG. 4. the subsidiary peak of particle $+$, although very small compared to the free fermion peak, can still be seen to be highly asymmetric both in the $x$-coordinate and rapidity space. In fact, it is more akin to a wave-front, with particles at the back slower than those in the front. As mentioned earlier, the effective velocity profile does not fully explain the decay and eventual disintegration of the peak. It is worth noticing that by time $t = 10$ the decaying peak is also significantly lagging behind the interacting peak of particle $-$ (since its propagating velocity is 9.4 whereas for the interacting peak it is $+1$) and eventually falls out of its interaction range.

- This is in stark contrast to the persistent peak in FIG. 5. Here, the peak is symmetric in the coordinate space and is accompanied by a velocity profile symmetric with respect to its maximum. The maximum of the subsidiary peak of particle $+$ coincides in the coordinate space with the maximum of the interacting peak of particle $-$ and both can be found at $x = 10$ for $t = 10$. Despite slower effective velocity, the propagation velocity of the subsidiary peak equals that of the magnet-like interacting peak, which interacts with the bath and drags the subsidiary peak along. This is the magnetic-fluid mechanism reported in the paper.

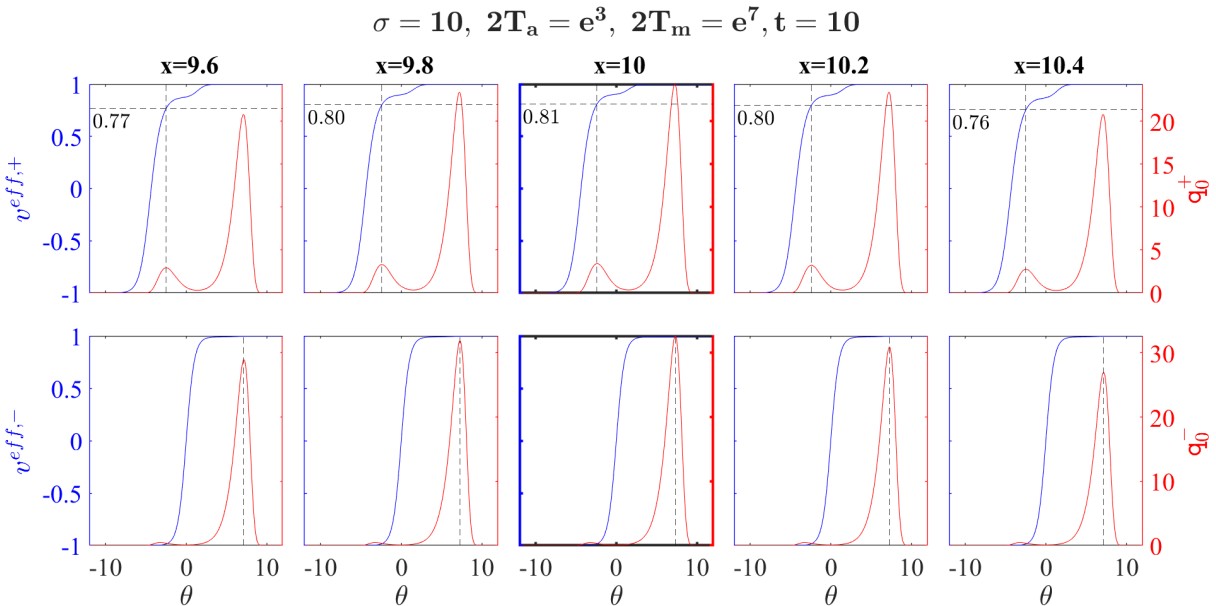

Figure 5: Effective velocities (left axis, red) and spectral density cross-sections (right axis, blue) of particle $+$ in row 1 and $-$ in row 2 at time $t = 10$ in the case of bath temperature $2T_a = e^3$. For particle $+$ we see the subsidiary (persisting) and free fermion peaks whereas for particle $-$ we see the (large) interacting peak a very small bump corresponding to the bath (for particle $+$ this is masked by the persisting peak sitting right on top). Columns show these quantities sequentially for positions $x = 9.6, 9.8, 10, 10.2, 10.4$. The emphasised central column corresponds to $x = x^* = 10$ the position of the maximum of subsidiary peak of particle $+$. The vertical dashed lines indicate the maximum of particle $+$ subsidiary peak ($\theta^* = -2.39$) in row 1 and the maximum of particle $-$ interacting peak in row 2, which is around the value $\theta^* + \sigma$ as dictated by the structure of the scattering phases. The horizontal dashed lines indicate the effective velocity of the maximum of particle $+$ subsidiary peak and are annotated with the corresponding numerical values.

# 5    Comparison with Free-Particle Evolution

Section 4 makes a compelling case for the crucial role that non-trivial interaction between particles $\pm$ plays in determining the dynamics of this model, giving rise to the distinct features that we have discussed in our paper: tail formation and the decay and/or persistence of localized particle densities.

An alternative way to look at these phenomena is to compare our results with results obtained in the absence interactions. We can then ask: what properties of the peak propagation, decay, tails and persistence are really characteristics of the formation and decay of unstable particles due to nontrivial interaction? The evolution of free particles from an initial distribution of velocities may produce tails due to dispersion. Is this enough to explain our observations?

In this section we show that this is not the case: the interaction is essential in order to explain the observed effects. Thus, the presence of interaction can be deduced from the results of the non-equilibrium dynamics. The striking difference between the HSG particle density and its freely-evolved version has already been shown in FIG. 1. Here we discuss these differences in more detail.

The comparison with free-particle evolution can be performed in a number of ways. One might consider the same quench problem, with the same initial temperature distribution, in a free theory. This however leads to a different initial density distribution, hence any comparison with the interacting case is not very meaningful. Instead, one may compare with what would happen for a free-particle system with the same initial spectral distribution of particles: the distribution in space-rapidity $\rho_p^\pm(x, 0, \theta)$, describing the density of particles with positions $x$ and relativistic velocities $\tanh\theta$ at time $t = 0$.

We note that both in the interacting and non-interacting systems, $\tanh\theta$ has the physical meaning of asymptotic velocity that would be observed if particles were let to expand an "astronomically large" time in the vacuum. A distribution in space-rapidity can be measured physically, in principle, by performing such an expansion independently for every mesoscopic cell in space. The time must be large enough for all particles in the cell to spatially separate, no matter how small their initial rapidity differences. This asymptotic distribution of rapidities, in each fluid cell, does not depend on the interaction; it is thus a universal characteristic, and the comparison between the interacting and non-interacting cases makes sense.

The comparison is presented in FIG. 6 (the first row of this figure is similar to row 2 of FIG.1), where we see stark differences between free and interacting particle evolution. In order to interpret the results, we remark, from FIG. 1 in the main text, that the distribution of rapidities is very different from that of the initial effective velocities: particles + with negative rapidities that pertain to the subsidiary peak have *positive* effective velocities, instead of negative. Hence, with the free dynamics, these particles will start moving towards the left instead of the right as observed with interactions. Thus, free-particle evolution according to the initial spectral distribution is significantly different from the outset, with peaks of very different sizes. Furthermore, the distribution of rapidities leads to sharply defined velocities, which are all either very near to 1 or to $-1$. Hence there is very little dispersion, and *no tail or decay is observed*. Those are thus a stark indication of the presence of interaction. It is also striking to observe that in the case without bath, the decay under interacting evolution leads to a final right-moving wave that agrees precisely with the right-moving wave emitted from the initial distribution in

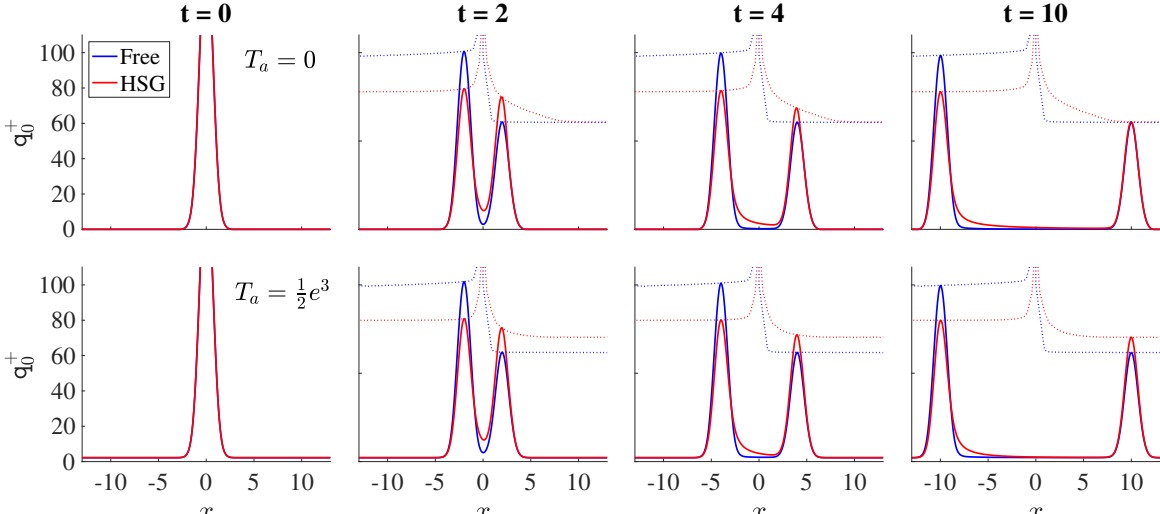

Figure 6: Particle density in the HSG-model $\rho_p^+(x,t,\theta)$ (red) versus $\rho_p^+(x - t\tanh\theta, 0, \theta)$ (blue). The top/bottom rows correspond to the absence/presence of a bath. The peak in the first column has been cut so as to show the remaining figures more clearly. The dotted curves represent the change in the height of the local maxima of the spectral densities as function of time. Red and blue curves generally differ in the height of the local maxima as well as the presence (absence) of a tail.

the free evolution case. This indicated that all unstable particles, represented by the original subsidiary peak, have decayed, and lie within the tail. With a bath, there remains an additional particle density, representing the population of unstable particles stabilised by the presence of the bath.

Another way of approaching these ideas is to consider what would happen if we were to consider the free-evolution problem with HSG initial conditions not only for the spectral density $\rho_p^\pm(x, 0, \theta)$, but also for the velocities $v^{\mathrm{eff},\pm}(x, 0, \theta)$: setting the initial velocities $\tanh\theta$ of the free particles at position $x$ and rapidity $\theta$, to the value of the effective velocity $v^{\mathrm{eff},\pm}(x, 0, \theta)$. We emphasise that this is more difficult to justify physically, as the effective velocity already encodes nontrivial aspects of the interaction. However, it makes the free and interacting problems closer, as both the initial condition, and its initial infinitesimal change, are the same. This allows us to separate any features that are solely dependent on the initial state from features that are truly dependent on the nature of the interaction in the model. This is interesting because the initial density and effective velocity profiles are non-trivial. In particular, there is a whole spectrum of effective velocities in the initial state, and so one may wonder to what extent the dispersion due to this spectrum of velocities explains the tails and decaying/persisting peaks.

In this case we numerically implemented the free evolution using a molecular dynamics simulation. The resulting particle density $q_0^+(x, t)$ is presented in FIG. 7 (in blue) and compared with the corresponding function for our model (in red). Because of the statistical nature of the simulation we see some noise around the blue curve. The main behaviours are however clear.

First, a tail is also present for free evolution due simply to the presence of particles with

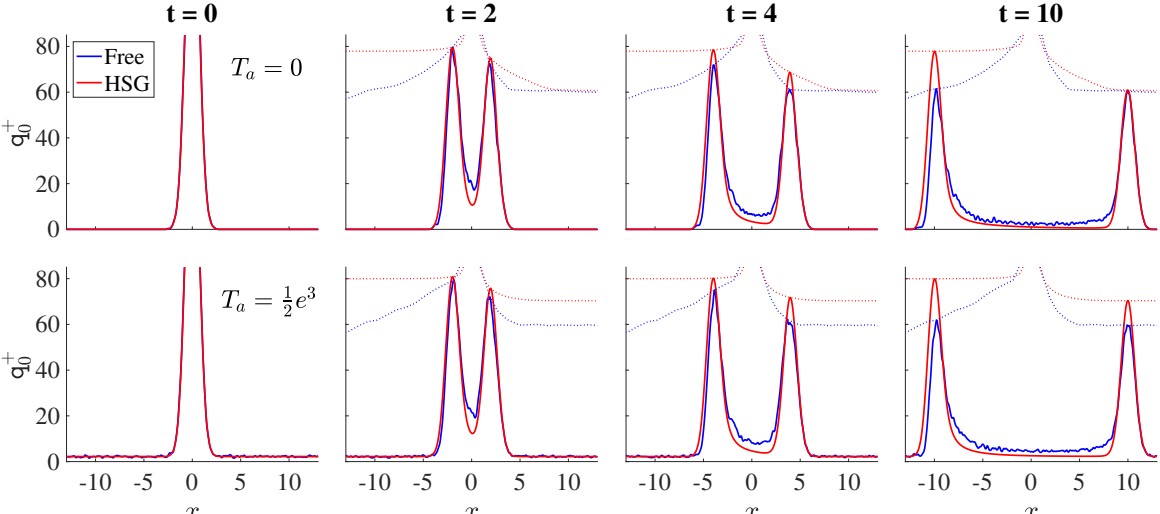

Figure 7: Particle density in the HSG-model (red) compared to the particle density that would result from free evolution of the same initial conditions (blue). The top/bottom rows correspond to the absence/presence of a bath. The peak in the first column has been cut so as to show the remaining figures more clearly. The dotted curves represent the change in the height of the local maxima of the spectral densities as function of time. Red and blue curves generally differ in the height of the local maxima as well as the shape of the tail.

intermediate velocities (e.g. neither 1 nor -1). However, the shape of the tail is noticeably different. Indeed it contains a much higher density of particles than in the interacting model. This is due to the fact that in the interacting model even slower particles are carried forward by their interaction with particles of the opposite type so that their propagation velocity is higher than their effective velocity (see Section 4) and the decaying process is slowed down. Essentially, this highlights the fact that unstable particles have a nonzero lifetime, and thus carry densities more coherently, limiting the dispersion due to the original velocity spectrum.

Second, in the absence of interaction the left peak decays in time for all times we have reached numerically (until, presumably, it eventually fully disintegrates, due to the slower particles contained in it), whereas the right peak experiences some decay only for early times. Instead, in the interacting model, the left peak experiences no reduction whereas there is some decay of the right peak which is however slower than for the free model.

Finally, in the HSG-model in the presence of a bath there is a magnetic effect that gives rise to a persistent particle density which contributes to the right peak of the particle density. This effect is clearly absent for free evolution and as a result the red right peak is higher than the blue peak for large times in the second row, whereas they are equal on the top row. The extra density in the interacting model is a signature of the presence of a persisting population of unstable particles, and is a specific characteristic of the interaction.

In summary, the signatures of instability, particle creation and decay that we have discussed in our paper are robust features of our model which are strongly determined by interaction and

are not found for other related scenarios involving free particle evolution.

# 6  Discussion and Conclusion

In this paper we have shown how unstable particles in integrable QFT can be seen in a new light within the GHD framework. The presence of an unstable excitation substantially alters the features of the densities and effective velocities of its stable constituents, an analysis of which shows that, after release into a zero- or low-temperature environment, unstable particles decay. In spatial density profiles, this is hallmarked by the slow decay of propagating waves and the formation of tails, which cannot be explained by dispersion from free-particle evolution. Importantly, in the presence of a bath, a new hydrodynamic phenomenon comes into play: the long-time persistence of a small but significant wave of unstable particles "riding" on top of the bath, propelled by its interaction with the large wave of particles of the opposite type.

We have chosen temperatures that are large enough so that the presence of a continuum of fluid modes in integrable systems does not play an important role. The physics is dominated by the renormalisation-group flow from interacting ($c = 6/5$) to free ($c = 1$) conformal field theory. Importantly, our results suggest that the decay of the population of unstable particles is a real-time dynamical implementation of this flow, an idea which has may lead to a deeper understanding of non-equilibrium dynamics. It is this flow that determines the required values of temperatures, and, as in our specific setup, they are achievable at the hydrodynamic scale. The distinct signatures of instability identified within GHD are explained via the combined phenomenology of unstable particles and hydrodynamics, and should reflect universal behaviours.

Given how rich the dynamics of this simple model is, it would be very interesting to study other models of the same HSG family where more unstable particles with tunable masses and decay widths are present. There is also still much to learn about the time evolution of hydrodynamic quantities under variation of $T_a, T_m$ and $\sigma$, which we will address in [54]. We hope that our work will suggest ways in which the decay of matter in closed many-body quantum systems may be experimentally identified.

*Acknowledgements.*— We are grateful to Frederik S. Møller for answering our questions on the iFluid package. In 2020 two groups of courageous colleagues put together the conference "Great Lessons from Exact Techniques and Beyond" held in Padova 20–25 September 2020 and the summer school on "Clean and disordered systems out of equilibrium" held in Cargèse 14–18 September 2020. We thank the organisers of both, for their financial support, for giving us the opportunity to meet in person and discuss and, especially, for creating islands of normality in the midst of a global pandemic.

# A   Numerics with iFluid

In this Appendix we describe in more detail the numerical simulations presented in the paper, focussing on possible error sources, precision issues and consistency checks.

## A.1   Generalities

As mentioned in the paper, our results have been obtained with iFluid (integrable-Fluid, version 1.1.0) which is an open-source MATLAB framework specifically designed for solving the GHD equations in integrable models [53]. It is worth mentioning also that an integral-equation solution to the GHD equation (8) is also known [48], but its numerical stability has not been sufficiently studied yet.

Thanks to a tensor-based numerical environment, iFluid boasts high efficiency and high process running speed. Additionally, a new model can be easily implemented by extending the iFluid package with a model-specific class. Such implementation can be done by following the instructions provided in [53]. However, the $SU(3)_2$-HSG model requires additional modifications and we have implemented them in this study.

The quantities studied in this paper feature in the `iFluidCore` class, which provides solutions to the TBA equations and inputs for the GHD equations. By default, the definition of spatial particle density included in the `iFluidCore` class, depends on the type-array. It is a fundamental property of the class and is included in the model implementation as a one-dimensional array composed of integers which label the particle types. In massive integrable QFTs with more than one particle in the spectrum, this definition of the particle density introduces multiplicative factors in its type components, which can lead to wrong results. We have instead implemented the definition that reproduces the total spatial particle density $q_0(x,t) = q_0^+(x,t) + q_0^-(x,t)$ as defined in the main body of the paper and leads to a correct result. Moreover, in our implementation of the model, some `iFluidCore` functions have been modified to output the contributions from each particle type to a given quantity separately (i.e. the functions $q_0^\pm(x,t)$).

In iFluid, the propagation of the GHD quantities is computed via `iFluidSolver` class. In our numerical simulation we have employed the `SecondOrderSolver` solver, whose details can be found in section 3.2 in [53].

To the best of our knowledge, our work provides the first application of the iFluid package to a system which is initially prepared in a state involving temperatures $T \gg 1$. Earlier examples provided with the package (i.e. sinh-Gordon model) were tested for temperatures $T \approx 1$. We have successfully ensured the convergence of the thermodynamic Bethe ansatz equations evaluated by the `iFluidCore` class for temperatures up to $\mathcal{O}(e^{10})$. We have checked the consistency of our results in several ways.

## A.2   Precision and Consistency Checks

In order to make sure that the modified code gave meaningful results we carried out various consistency checks, mainly comparing the outputs of iFluid with standard results that are accessible by other numerical procedures.

A preliminary check was done on the initial state, which is given by the solution of the (equilibrium) TBA equations for a given fixed temperature $T(x)$ for each value of $x$. In the iFluid code, the precision is controlled by two parameters, namely, the tolerance and the maximal number of iterations allowed. In order to guarantee the highest accuracy, we set the former to $10^{-32}$ and the latter to 5000. We made these choices in part by comparing the outputs of iFluid in the initial state to results obtained for the same functions with a Mathematica code used in [32], and established that, for the choices above, we achieved higher precision with iFluid.

A similar check was performed by evaluating the TBA scaling function $c(T)$ over a range of temperatures (especially at high temperatures), and seeing that plateaus at the expected values of the central charge [41] where reproduced. Likewise we computed the energy densities and currents in the UV (high temperature) limit, reproducing once more the results of [32].

| $t$ | 0 | 3 | 6 | 9 | 12 | 15 |
|-----|---|---|---|---|----|----|
| $Q_0$ | 561.2520 | 561.2541 | 561.1937 | 561.1511 | 561.1156 | 561.0970 |

Table 2: Numerical values of the total particle density $Q_0$ at several times. Here, the parameters are: $\sigma = 10$, $\log(2T_m) = 7$ and $T_a = 0$ (no bath). To evaluate $Q_0$, we have computed (4) in the main paper and performed a cubic spline interpolation, implementing additional grid points whose spacing in $x$ is $\Delta x = 1. \times 10^{-5}$. $Q_0$ is numerically conserved, up to a variation on the first decimal place, which can be attributed to the discretisation procedure.

Having established that the ground state is accurately described, we then turned to consistency checks of dynamical quantities. We calculated the total particle density $Q_0$ given by the $x$-integral of the sum of partial densities in Eq.(4) of the main paper. $Q_0$ should be conserved in time and so its computation for various values of times provides a consistency check for numerical solutions of the GHD equations. In Table 1 we have reported the numerical values of the total particle density evaluated in the no-bath case. $Q_0$ is confirmed to be conserved, up to a numerical variation on the first decimal place, which is the order of the grid spacing implemented in the simulation (see Table 2 for details).

Finally, we performed other consistency checks which exploit the connection of our model with free theories. In particular, as explained in the main body of the paper, we have that for $T \ll e^{|\sigma|/2}$ our model should reduce to two Majorana free fermions. Thus, performing numerics for the $SU(3)_2$-HSG model with $\sigma = 20$ for the same temperature choices discussed in this paper, we should obtain results which are fully in the free fermion regime where the GHD equations can be solved exactly. Thus, in this regime numerical results from iFluid can be compared to analytical solutions. We have confirmed that they are in perfect agreement.

## A.3 Space and Rapidity Discretisations

Besides convergence of the numerical solution of the TBA equations, we have established that the key source of numerical error is the choice of space discretisation. The values adopted in our computations are given in Table 3.

We have devoted special attention to the discretisation of the rapidity interval. iFluid employs Gauss-Legendre quadrature integration, which has excellent convergence properties for integrals over a finite interval [55, 56]. However, this quadrature is optimised for minimising

| | $\Delta_{\mathbf{max}}$ | No. Points | Max. Val. | Quadrature |
|---|---|---|---|---|
| $t$ | 0.2500 | 61 | 15. | Rectangular |
| $x$ | 0.1000 | 441 | 22. | Rectangular |
| $\theta$ | 0.0762 | 700 | 17. | Gauss-Legendre |

Table 3: Discretisation parameters and quadratures chosen for the two numerical simulations discussed in this paper (i.e. with and without a bath). From the left, the columns indicate (respectively for $t, x, \theta$): the variable, the lowest resolution (i.e. the largest spacing between two grid points $\Delta_{\mathbf{max}}$), the number of grid points, the largest absolute value the variable takes, and the type of quadrature implemented. In the rectangular quadrature we have implemented equidistant grid points with fixed spacing $\Delta_{\mathbf{max}}$.

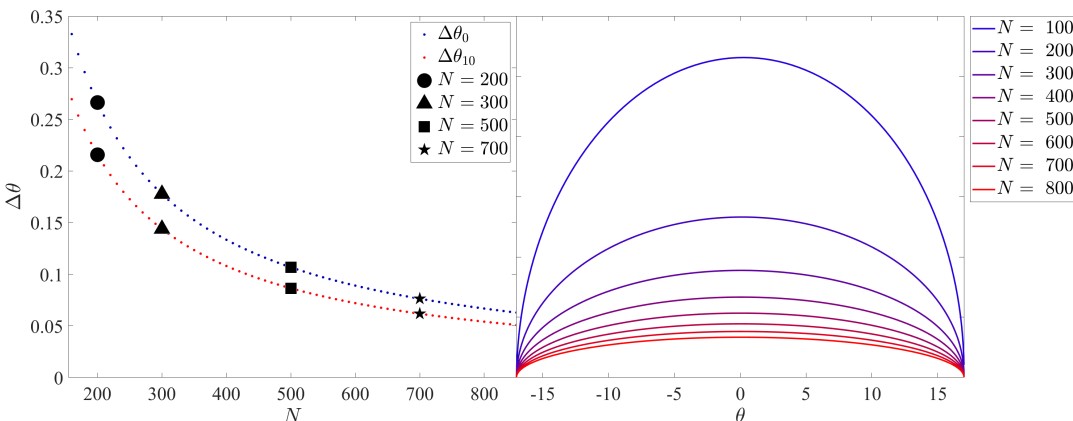

Figure 8: Right: The rapidity discretisation interval as a function of rapidity for different numbers of points in a fixed rapidity interval $\theta \in [-17, 17]$. The markers indicate the values of $N$ that are considered in FIG. 9. Left: Comparison of the rapidity discretisation interval in the middle and towards the edges of the interval with non-trivial dynamics. $\Delta\theta_0$ and $\Delta\theta_{10}$ are defined to be the rapidity discretisation intervals for $\theta = 0$ and $\theta = 10$, respectively.

boundary errors. This means that the number of data intervals is lowest in the middle of the interval, precisely where we find the non-trivial behaviour of the thermodynamic quantities of interest. The dependence of the size of the rapidity intervals, $\Delta\theta$ on the rapidity $\theta$ and the number of points in the interval considered $N$ for a fixed range of rapidities is shown in FIG. 8.

As a point of reference, we have chosen to compare the resolution at $\theta = 0$ and 10, as a primary region with non-trivial dynamics. Although the discretisation interval for $\theta = 0$ remains the largest for any value of $N$, the distribution quickly flattens out when the number of points is increased. Investigating the range of $N$ values from 200 to 700, we achieve an order of magnitude increase in resolution for both $\theta = 0$ and 10. Significantly smaller is also the difference between these two quantities, which is consistent with the flattening of the curve in the right panel of FIG. 8. The numerical values of the size of the discretisation interval for a chosen numbers of points in the rapidity interval are included in Table 3.

| $N$ | $\Delta\theta_0$ | $\Delta\theta_{10}$ | $\Delta\theta_0 - \Delta\theta_{10}$ |
|---|---|---|---|
| 200 | 0.2664 | 0.2157 | 0.0506 |
| 300 | 0.1777 | 0.1439 | 0.0338 |
| 500 | 0.1067 | 0.0864 | 0.0203 |
| 700 | 0.0762 | 0.0617 | 0.0145 |

Table 4: Numerical values of the rapidity discretisation intervals $\Delta\theta$, for a given number of rapidity points $N$ in the range of rapidities used in this investigation $\theta \in [-17, 17]$.

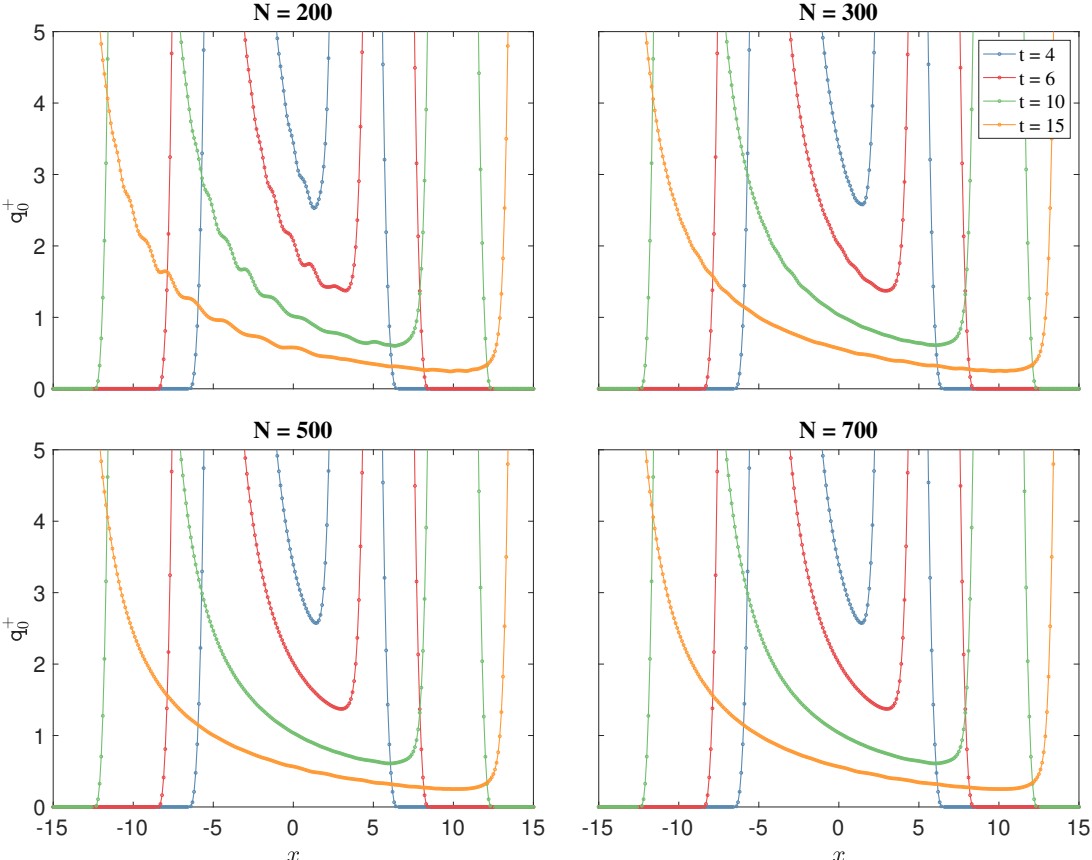

Figure 9: Spatial particle density of particle $+$ for different numbers of points in the fixed rapidity interval $\theta \in [-17, 17]$. The $N = 200$ panel shows regular oscillations. The oscillations persist for $N = 300$, disappear around $N = 500$ and are consistently absent for higher numbers of points, such as $N = 700$. Variations of space and time discretisation parameters have not produced any significant changes in the same functions.

In order to obtain the averages of conserved charges and currents in the $SU(3)_2$-HSG model, it is necessary to integrate over the rapidity variable, as defined in (4) of the paper. This integration procedure leads to the "accumulation" of any numerical errors present in the original

function and to the formation of regular structures that could be easily mistaken for genuine physical phenomena. An example of this effect can be seen in the particle density associated to particle + computed with different rapidity discretisations. Four examples are presented in FIG. 9. As expected, the problem arises from the middle of the rapidity interval where resolution is lowest. By changing $N$ we can identify a large enough value that guarantees a stable solution for the spatial particle density. In our code we have chosen $N = 700$. The quantities not integrated over the rapidity integral did not exhibit any unusual behaviour even for the values of $N$ as low as 200. Interestingly, this applies also to the spectral density, which produces then spatial particle density when integrated over rapidity. This further confirms that the emergence of of the oscillations seen in the top left FIG. 9 is genuinely a numerical effect arising due to integration.

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
