# Peer review of "Tails of Instability and Decay: a Hydrodynamic Perspective"

_SciPost Physics_

## Round 1 · Referee Report · Anonymous (Referee 1) · 2021-10-1

Report

This paper applies the generalized hydrodynamics formalism in the somewhat unusual setting of integrable field theories with unstable excitations. The specific problem was first studied by some of the authors themselves in Ref. [32], which the present work extends to the problem of “free expansion” into a low-temperature bulk state. I find the authors’ results physically interesting and thoroughly interpreted, and a solid contribution to understanding the consequences of unstable excitations far from equilibrium. However, I request that the authors clarify the following points before the manuscript is published in SciPost:

Fig. 1: this figure gets across the central message of the paper, that particle decay leads to unusual “tails” of the sound modes. While I understand that the initial state is prepared using TBA with parity-breaking phase shifts, I find it a little confusing that the free evolution breaks parity. For example, in a generic non-driven physical system, a thermal state of the form (2) would be parity even. Is there any sense in which parity even/odd observables are more physical or natural in the QFT considered by the authors? (e.g. the sum q_0^+ + q_0^-) Do such observables show the same phenomenology as Fig. 1? And are there parity conserving models with similar physics?

p6: the bath temperature seems to be restricted so that the bath is “free”. Could the authors comment on what changes when the bath is interacting? (i.e. T_a > e^{\sigma/2}). There also seems to be a typo (missing log/exp) in the inequalities for T_a and T_m at the bottom of p6.

p6: “This is akin to having a fluid that is magnetic, and running a magnet pass it. We see a wave that follows the magnet but the fluid itself does not need to move.” The intuition makes sense but the hydrodynamics of magnetic fluids (MHD) is complicated. Could the authors add a specific reference, or a calculation, for the phenomenon they have in mind? This would be helpful because the magnetic analogy is mentioned several more times.

p14: “the decay of matter in closed many-body quantum systems may be experimentally identified”. This sounds very interesting, but which experimental systems do the authors have in mind? As mentioned above, parity violation seems a little unlikely for the initial state (2) in a closed quantum system. Do the authors expect the tail in the sound mode to appear more generally? Or is there an effective experimental Hamiltonian that realizes the specific type of model (parity breaking, relativistic, integrable, unstable excitations) studied in this paper?

A2, Table 2: Since the authors are being thorough with the simulation details, one should also check the conservation of higher conserved charges that sample more of the UV. Is the accuracy similarly good for the charge directly relevant to the initial condition, i.e. energy?

A3: I did not understand how the error benchmarking was done in this appendix – Fig. 9 shows convergence but relative to what? It might clarify matters to add some measure of the error, as in Table 2.

Finally, some minor comments on presentation:

Notation for excitations: using + and – in the main text was a little confusing, as each time I read them as a connective. Could the authors consider writing e.g. (+) or (-), or using some other specialized symbol?

p2: “generalised hydrodynamics (GHD) [26, 27], the hydrodynamic theory which accounts for generalized thermalisation”: GHD is indeed a very successful theory that pins down thermalization to GGEs in certain contexts (when the hydrodynamic assumption holds). But it is not expected to account for all types of approach to GGE (e.g. violent quenches, small systems…).

p2: The GHD necessitates -> GHD requires?

p3: “excitations mostly move at velocities \pm 1”: a qualifier like “relativistic” should be added, since this is not true in all free field theories.

p5: “a spectrum effective velocities” -> spectrum of

p6: “magnet pass” -> magnet past

p14: “an idea which has may lead to a deeper understanding”: has or may?

p19: of of -> of
  • validity: -
  • significance: -
  • originality: -
  • clarity: -
  • formatting: -
  • grammar: -

Author:  Olalla Castro-Alvaredo  on 2021-12-01  [id 1994]

(in reply to Report 1 on 2021-10-01)
Category:
remark
answer to question
reply to objection
correction

We thank the referee for their detail comments and careful reading of the paper. Below we address their comments and explain how and where we have modified the paper.

Comment 1: Fig. 1: this figure gets across the central message of the paper, that particle decay leads to unusual “tails” of the sound modes. While I understand that the initial state is prepared using TBA with parity-breaking phase shifts, I find it a little confusing that the free evolution breaks parity. For example, in a generic non-driven physical system, a thermal state of the form (2) would be parity even. Is there any sense in which parity even/odd observables are more physical or natural in the QFT considered by the authors? (e.g. the sum q_0^+ + q_0^-) Do such observables show the same phenomenology as Fig. 1? And are there parity conserving models with similar physics?

Answer:
Regarding the issue of free evolution, the reason why parity breaking is observed is because the initial state breaks parity. In other words, what we are doing in this section is taking the HSG initial state functions (e.g. effective velocities, spectral densities at t=0) and evolving them freely. So, the parity breaking is incorporated into the choice of initial state. We agree that in a generic system for a parity symmetric initial state this would not be observed. Our point in this section was to show that the features observed in our model could not be fully explained by the choice of initial state only. The same initial state freely evolved does not produce the same long-time outcomes (such as for instance the formation of a persisting peak). This means that the non-trivial interaction in the model, leading to the formation of unstable particles, plays an important role in explaining the dynamics that we see.

Another observation, which is more general for our model is that the parity breaking that is seen here between the + and – quantities is a consequence of considering those quantities separately. This is somewhat unusual as one tends to consider total densities (such as q_0^+ + q_0^-) rather than the individual contributions and the same applies to higher spin conserved quantities. In our work on this model, we have generally considered the (+) and (–) contributions separately, because from our definition, these quantities are separately conserved (as they are linear in the spectral density) and also because these quantities reveal the distinct physics of the problem is a clearer way. If we were to plot q_0=q_0^+ + q_0^- we would also see decay and tails, it is just that we would see them both on the right and on the left, in a parity symmetric fashion (as can be easily imagined by adding up the functions in the middle row of Fig. 1 and their parity reflected versions in the x-variable). However, for instance, the presence of a free fermion peak would be obscured if the two contributions are added up, which is a rather interesting feature which relates directly to the structure of the scattering phases. We have added a comment after equation (5) to clarify this.

We do not know of any parity conserving models where exactly the same kind of physics is recovered, but this is mainly because there are very few known integrable theories which incorporate unstable excitations. The most similar example to compare with would be the staircase model that was studied recently in the paper arXiv:2105.13349 (reference [56] in our new version). This theory has an S-matrix which also depends on a free parameter and tuning that parameter also leads to GHD functions that are rather peculiar. This S-matrix does not break parity and there is only one type of stable particle. Although the physics of this model is not usually described in terms of the formation of unstable particles, looking at its c-function at thermal equilibrium, for instance, the staircase pattern observed is compatible with the interpretation that, above some energy threshold, an unstable particle is formed, which, unlike in the HSG model, for sufficiently high temperatures, can bound again with itself to produce another more massive unstable particle, and this process of formation of ever heavier unstable excitations continues up to the UV limit of the theory. We have added a paragraph in the conclusion referring to this work which was not cited in the original version (as it came out later).

Comment 2: p6: the bath temperature seems to be restricted so that the bath is “free”. Could the authors comment on what changes when the bath is interacting? (i.e. T_a > e^{\sigma/2}). There also seems to be a typo (missing log/exp) in the inequalities for T_a and T_m at the bottom of p6.

Answer: We have corrected the typo, thanks!
Regarding the regime, the referee is right that we have focussed on a situation where the bath is free. When the bath is interacting, similar effects are seen (in fact, we are just about to finish a new preprint where we consider some of these effects). However, when there is interaction throughout, it is a bit harder to isolate evidence of particle decay/persistence.

Our main aim was seeing signatures of decay as clearly as possible. It is natural to expect that if in the initial state we have a high concentration of unstable particles around the origin and then we release them into a cold environment, they will start decaying rather fast and we should see some evidence of that (as we actually do). If the bath is not very cold but colder than the region around the origin, decay is still seen, but this is slower, the smaller the temperature gradient is and would force us to consider much larger simulation times to gain similar insights.
It is also interesting to think of the problem in RG terms. The main peculiarity of our theory is that, depending on energy scales, the UV fixed point is either two free fermions or a non-trivial CFT of central charge 1.2. Our choice of temperatures allows us to explore the two regimes simultaneously (although our temperature T_m always remains a little below the deep UV!)

Comment 3: p6: “This is akin to having a fluid that is magnetic, and running a magnet pass it. We see a wave that follows the magnet but the fluid itself does not need to move.” The intuition makes sense but the hydrodynamics of magnetic fluids (MHD) is complicated. Could the authors add a specific reference, or a calculation, for the phenomenon they have in mind? This would be helpful because the magnetic analogy is mentioned several more times.

Answer: We are sorry that our description is confusing. In fact, our use of the term “magnetic fluid” is meant in a more superficial sense than it perhaps sounded. Indeed, the physics of magnetic fluids is a whole other field of discussion, and we did not intend to claim we see such physics in our model. For instance, interaction between particles of the same type is trivial in our model, they don’t interact, which means that the usual self-interaction effects that one finds in magnetic fluids (due to the formation of dipoles that then interact with each other) is totally absent. We just meant that the phenomenology we observe looks qualitatively similar to the sort of interaction that takes place when a magnet is run by a magnetic substance. We have added a footnote on page 6, where the reference to a magnetic effect appears for the first time, to emphasize this.

Comment 4: p14: “the decay of matter in closed many-body quantum systems may be experimentally identified”. This sounds very interesting, but which experimental systems do the authors have in mind? As mentioned above, parity violation seems a little unlikely for the initial state (2) in a closed quantum system. Do the authors expect the tail in the sound mode to appear more generally? Or is there an effective experimental Hamiltonian that realizes the specific type of model (parity breaking, relativistic, integrable, unstable excitations) studied in this paper?

Answer: Our comment here is speculative rather than precise. We do not know how a comparable system may be realized experimentally and we agree with the referee that most examples of parity breaking, for instance in spin chains, occur either for open or non-Hermitian systems. However, it is not necessarily the case that parity breaking will always accompany the presence of unstable modes and their decay, even if it is a feature of this very particular model. The essential idea is that particle densities and even spectral densities can nowadays be measured in cold atom experiments (for instance, when a density of cold atoms is first confined by some trapping potential and then released and allowed to expand). See for instance our reference [23]. So it is possible that experiments might reveal systems where for instance these densities exhibit tails in which case our work would suggest there may be unstable excitations playing a role. The idea is that our work provides a toolbox of possible behaviours that haven’t been seen before and then maybe one day, an experimental set-up will reveal an application of our results. We modified a little bit the relevant sentence in the conclusion.

Comment 5: A2, Table 2: Since the authors are being thorough with the simulation details, one should also check the conservation of higher conserved charges that sample more of the UV. Is the accuracy similarly good for the charge directly relevant to the initial condition, i.e. energy?

Answer: Yes, we have done extensive consistency checks in order to be satisfied that our GHD description was actually accurate, including for the energy and the entropy. In both cases we observed conservation with similar accuracy as for the quantity in table 2. We decided not to discuss those in the paper, as we were focussing on other quantities, but we have now added a comment in Appendix A, in the caption of Table 2.

Comment 6: A3: I did not understand how the error benchmarking was done in this appendix – Fig. 9 shows convergence but relative to what? It might clarify matters to add some measure of the error, as in Table 2.

Answer: In Fig. 9 what we are looking at are close-ups of the function q_0^+ (the same function as in Fig. 1, middle row) around x=0. The colours represent different times as indicated in the legend. What we are trying to show here is that, if the rapidity variable is not discretized finely enough, then unphysical oscillations can appear, as we can see in some of the figures. Since the rapidities are taken between -17 and 17, so in an interval of size 34, the rapidity differential is 34/N and so is smaller for larger N. Thus the convergence we refer to is relative to N. Convergence is achieved for large enough N and the figures show graphically what “large enough” means (N>=500). The value of this number plays a crucial role in the numerics because the integration in theta is carried out as a Riemann sum by iFluid. We have modified the caption of this figure a little bit, adding some useful details.

Finally, some minor comments on presentation:

Comment 7: Notation for excitations: using + and – in the main text was a little confusing, as each time I read them as a connective. Could the authors consider writing e.g. (+) or (-), or using some other specialized symbol?

Answer: We have now used (+) and (-) as suggested.

Comment 8: p2: “generalised hydrodynamics (GHD) [26, 27], the hydrodynamic theory which accounts for generalized thermalisation”: GHD is indeed a very successful theory that pins down thermalization to GGEs in certain contexts (when the hydrodynamic assumption holds). But it is not expected to account for all types of approach to GGE (e.g. violent quenches, small systems…).

Answer: We agree with the referee. The sentence was written in a misleading way. What we meant to say is that GHD is based on (or uses) generalized thermalisation (instead of “standard” thermalisation). We have modified the sentence accordingly.

Comment 9: p2: The GHD necessitates -> GHD requires?

Answer: Done.

Comment 10: p3: “excitations mostly move at velocities \pm 1”: a qualifier like “relativistic” should be added, since this is not true in all free field theories.

Answer: We have added the word “relativistic” in the sentence right after equation (7).

Comment 11: p5: “a spectrum effective velocities” -> spectrum of

Answer: Corrected.

Comment 11: p6: “magnet pass” -> magnet past

Answer: Corrected.

Comment 12: p14: “an idea which has may lead to a deeper understanding”: has or may?

Answer: Corrected.

Comment 13: p19: of of -> of

Answer: Corrected.

---

## Round 1 · Referee Report · Anonymous (Referee 2) · 2021-10-21

Strengths

1- Studying an integrable QFT with unstable particles, in a solvable model, could shed light on more generic field theories. 2-Some of the phenomena reported by the authors are qualitatively new in the context of GHD.

Weaknesses

1- The model is a bit exotic 2- Most of the results are obtained from iFluid, and the discussion is entirely qualitative.

Report

In this paper, the authors apply the generalized hydrodynamics framework to unusual integrable quantum field theories with unstable excitations. This builds on some of the authors' previous work. They focus on a specific integrable QFT, which is admittedly exotic, but admits unstable excitations which the authors argue have clear signatures in non-equilibrium dynamics. The results are primarily numerical, using the library iFluid, with some qualitative interpretation of the results. They uncover several new features (in GHD) due to unstable excitations, including unusual "tails" around the sound modes. A more thorough analysis with some analytical understanding would have made the paper a lot stronger, especially if the authors could draw some general conclusions for other, possibly non-integrable QFTs, but the paper is of sufficient quality for Scipost.

Some sentences in the text should be clarified or edited. For example, the last concluding sentence of the abstract "Our study shows that the out-of-equilibrium dynamics of many-body systems can be strongly affected not only by the spectrum, but also by excitations with finite life-times." sounds completely obvious to me. Excitations with finite life time are not uncommon in nature, and it is surely no surprise that such excitations can contribute to dynamics. Maybe the authors mean something else and should be more specific here. Other sentences like this one can be found in the manuscript, like in the conclusion: "We hope that our work will suggest ways in which the decay of matter in closed many-body quantum systems may be experimentally identified.". Observing the decay of excitation does not sound like a particularly new or exciting endeavor, so again this sentence should be clarified. What are the general conclusions from this specific theory that the authors believe carry over to more generic QFTs?
  • validity: high
  • significance: ok
  • originality: ok
  • clarity: good
  • formatting: excellent
  • grammar: excellent

Author:  Olalla Castro-Alvaredo  on 2021-12-05  [id 2007]

(in reply to Report 2 on 2021-10-21)

We thank the referee for their comments.

Comment 1: Some sentences in the text should be clarified or edited. For example, the last concluding sentence of the abstract "Our study shows that the out-of-equilibrium dynamics of many-body systems can be strongly affected not only by the spectrum, but also by excitations with finite life-times." sounds completely obvious to me.

Answer: It is a bit unclear to us why this would be obvious. This is only obvious if taken completely out of context and seen as a completely general statement. Of course, everything affects the dynamics, but we claim that the effect is strong, and in the out-of-equilibrium dynamics, something which we believe has not been addressed before. Many colleagues working in similar problems in our area, are mainly familiar with models where all excitation are stable, parity is preserved and scattering phases are peaked around zero rapidity. So we think, as obvious as it might seem to the referee, this sentence makes an important point that is not clear to everybody likely to take an interest in our work. The sentence the referee refers to is just in the abstract where shortness is of the essence. Its full meaning becomes clear when considering the rest of the paper, as is usually the case with abstracts.

Comment 2: Excitations with finite life time are not uncommon in nature, and it is surely no surprise that such excitations can contribute to dynamics. Maybe the authors mean something else and should be more specific here.

Answer: We strongly believe there is no need to be more specific here. What we mean has been explained repeatedly in the paper. To summarise once more: what is unusual are not the unstable excitations, but the whole package of observations that come with them, and that have not been made before. In particular: the fact that (1) they arise in an integrable model, (2) their effects are captured by a hydrodynamic theory, GHD, and (3) they lead to significant changes in spectral densities and effective velocities. Again, as far as we are aware, the effects of unstable excitations, out of equilibrium, in finite density states, on the hydrodynamics of a many-body quantum systems has not been addressed before. As mentioned above, most colleagues in the field would be unfamiliar with such examples and the fact that one may explore such rich phenomenology without abandoning integrability.

Comment 3: Other sentences like this one can be found in the manuscript, like in the conclusion: "We hope that our work will suggest ways in which the decay of matter in closed many-body quantum systems may be experimentally identified.". Observing the decay of excitations does not sound like a particularly new or exciting endeavor, so again this sentence should be clarified.

Answer: If the referee is aware of any experimental realizations of integrable models where decay is observed, we would be glad to hear about them. So far, we are not aware of any. We are also not aware of any experiments where decay in an out-of-equilibrium, finite density many-body quantum system, even a non-integrable one, has been measured by measuring spectral or particle densities and identifying the sort of signatures of decay that we see here. Also in response to other referees' comments, we have modified the conclusion slightly to make this more explicit.

Comment 4: What are the general conclusions from this specific theory that the authors believe carry over to more generic QFTs?

Answer: In the presence of unstable particles, particle densities and spectral densities develop tails and additional long-lived peaks. Such densities can now be measured in certain experiments and so it might be possible to see such signatures in experimentally realizable models and to connect them to the presence of unstable modes. We have modified this sentence in the conclusion as the first referee had a similar query. We hope it is improved now.

Since this has been observed by another referee too, let us make a comment on the “exotic” nature of the model. The model has unusual properties, like incorporating unstable excitations, but it is part of the large family of Toda field theories and as such, it is not particularly exotic. It has a well-known Lagrangian description and has been studied in enormous detail in the 90s, both at classical and quantum level.

Finally, regarding another comment by the referee, “A more thorough analysis with some analytical understanding would have made the paper a lot stronger”: First, the referee should note that there is a strong analytic underpinning to our analysis: GHD. We are not performing a numerical simulation of the model (which would be forbiddingly difficult), but instead we are solving the GHD equations numerically. Doing so, we are uncovering a phenomenology, using this analytical framework, which, as far as we are aware, would be impossible to observe by any other currently known analytical or numerical technique.

Of course, within GHD or older techniques such as TBA, analytical results are hard to obtain in a theory with non-trivial scattering such as this and so it is standard that many studies in this field are mainly numerical. We don’t think our paper falls particularly short in that respect. On the contrary, the adaptation of iFluid for the study of this model is in itself a useful contribution that has helped improve the code and inspired its developers to introduce various improvements. We are working on an extension of this work which will provide a better qualitative and quantitative understanding of these and additional results. We think including such results in the current paper would have made it extremely long and hard to read so we have always aimed for a two-part project with this paper identifying and highlighting the main signatures of instability and the sequel isolating those signatures and investigating them further.

---

## Round 1 · Referee Report · Anonymous (Referee 3) · 2021-11-16

Strengths

Well written

Weaknesses

Model is obscure and it is unclear if results have and generality.

Report

In the manuscript, “Tails of Instability and Decay: A Hydrodynamic Perspective”, the authors consider quantum quenches in the inhomogeneous $SU(3)_2$-homogeneous sine-Gordon model. The interest in this model arises because i) it has a relatively simple spectrum, consisting of two particle species, and ii) its two particle S-matrix has a pole away from the imaginary rapidity axis indicating an unstable bound state or resonance. These two facts allow it to be studied with GHD and to observe the effects of bound state formation and decay in a non-equilibrium setting.

The authors study a quench where the system is initiated with a temperature profile hot in the middle (and so with a high density of particles), cooling off and becoming sparser as one moves away from the center. We see that because of the presence of an unstable bound state, the initial particle distribution of a given particle species in $\theta$ has three peaks, two main peaks directly correlated to the temperature of the gas and one smaller one related to unstable bound state formation. Because of the parameters chosen for the quench, the two main peaks are in the ultra-relativistic limit and move with the speed of light retaining their shape as the gas is allowed to evolve in time. The third peak however gets smeared out, a consequence of bound states forming and breaking apart. If the initial temperature profile of the gas is such that the gas asymptotes to a finite temperature (i.e. $T_a>0$) as one moves away from the center, this third peak initially gets smeared but retains its integrity unlike the $T_a=0$ case where the subsidiary peak disappears at late times.

Major critique:

The authors are motivated to the study this model because of the presence of unstable bound states in a model that is nonetheless integrable and so allows them to employ the framework of generalized hydrodynamics to study its non-equilibrium behavior. The model that they study this phenomenon in is, in the words of another referee, exotic. I do not think this model has any intrinsic interest in and of itself. There are, to the best of my knowledge, no material realizations, even in cold atomic settings, of homogeneous sine-Gordon models. (If there are, the authors need to say as much.) This leaves a rationale where one might hope to get some insight in general in theories with unstable particles by studying the phenomenon here.

However, I doubt quenches in this model can provide any real insight into a generic non-integrable models with unstable bound states. While bound state formation might be unstable, it is still special inasmuch as the dynamics in the model occur pairwise. Such pairwise interactions (i.e. that a n-particle S-matrix can be understood solely through a two-particle S-matrix) are very particular to an integrable model. In a non-integrable model, the interactions of a dense n-particle system won’t have a description as a product of two-body resonances but will be something much more complicated.

This leaves me unsure on what to recommend for this manuscript. It is certainly competently executed. However, I question the intrinsic interest here because the model is obscure and because I do not believe that the non-equilibrium dynamics here shine light on the case of unstable bound states in non-integrable models.

To bring it over the line for publishability, I think the authors have to address these concerns. Perhaps they could make the case that I am being unfair in thinking that homogeneous sine-Gordon models are non-generic. (It would be, for example, nice in this regard to know that these models have a Hamiltonian or Lagrangian formulation even if it exists as the scaling limit of a lattice representation.) It would also be good to understand when the analysis herein might be applicable in a non-integrable setting. Could a model with weak integrability breaking where the bound states pick up a finite lifetime lead to phenomenon like that seen here? And if so, under what conditions?
  • validity: high
  • significance: low
  • originality: ok
  • clarity: good
  • formatting: excellent
  • grammar: excellent

Author:  Olalla Castro-Alvaredo  on 2021-12-05  [id 2008]

(in reply to Report 3 on 2021-11-16)

Comment 1: "I do not think this model has any intrinsic interest in and of itself"

Answer: We do not see any scientific argument from the referee to justify their point. The idea that a model needs to be realized in a cold atom experiment to be of interest seems narrow minded to say the least, and the range of publications in SciPost, both theoretical and experimental, demonstrates this.

Comment 2: "I doubt quenches in this model can provide any real insight into a generic non-integrable models with unstable bound states", "I do not believe that the non-equilibrium dynamics here shine light on the case of unstable bound states in non-integrable models", " It would also be good to understand when the analysis herein might be applicable in a non-integrable setting. Could a model with weak integrability breaking where the bound states pick up a finite lifetime lead to phenomenon like that seen here? And if so, under what conditions?"

Answer: We believe the insights we obtain here are interesting already in the context of integrable models, and are worthy of publication on that basis. In addition, many phenomena in integrable models give insight beyond integrablity. In particular, even if scattering involves many-body processes, scattering channels can exist where unstable particles can form, which is what is studied here. Further, as usual, having a simple model like this one allows us to capture the essence of the phenomenon in a clear way. Finally, as the referee must be aware, there is a very large body of literature in which integrable models are seen to be relevant in experimental settings, where integrability is obviously broken. Under weak integrability breaking, for a long experimental time, dominant effects in one dimension are two-body effects, where integrability, including GHD, are fully applicable; see for instance the quantum Newton cradle experiment, other more recent experiments, and their GHD description (see the citations in the paper). We do not believe there is more to say than what we have already said in the introduction of our paper on this aspect, as going into any detail of this is beyond the scope of our paper. Nonetheless, because referees 1 and 2 also had questions about experimental applicability, we have slightly amended the conclusion to make our claims more precise.

Comment 3: "because the model is obscure", "they could make the case that I am being unfair in thinking that homogeneous sine-Gordon models are non-generic. (It would be, for example, nice in this regard to know that these models have a Hamiltonian or Lagrangian formulation even if it exists as the scaling limit of a lattice representation.)"

Answer: We do not know what is meant by “obscure” (is it only based on the referee's estimate of the number of people who have studied it until now?), and we do not believe the referee's assessment of the model as obscure is relevant.

In fact this model is part of the best studied family of integrable models there is: Toda field theories. As such, it is not obscure at all. There is a large body of literature which we cited in our paper in which this and other models of the same family have been formulated with an explicit Lagrangian description (they are perturbations of Wess-Zumino-Witten). Their classical and quantum integrability have been shown, their particle spectrum obtained semiclassically and finally their exact S-matrices obtained. It was not the aim of our paper to dwell on these old results (from the 90s) but all the relevant literature is cited there.

---

## Editorial Decision

resubmitted